# Learning knot invariants across dimensions

**Jessica Craven**[a] **, Mark Hughes**[b] **, Vishnu Jejjala**[a] **, Arjun Kar**[c]

[a]*Mandelstam Institute for Theoretical Physics, School of Physics, NITheCS, and CoE-MaSS,*
  *University of the Witwatersrand, 1 Jan Smuts Avenue, Johannesburg, WITS 2050, South Africa*

[b]*Department of Mathematics, Brigham Young University,*
  *275 TMCB, Provo, UT 84602, USA*

[c]*Department of Physics and Astronomy, University of British Columbia,*
  *6224 Agricultural Road, Vancouver, BC V6T 1Z1, Canada*

  *E-mail:* jessica.craven1@students.wits.ac.za,
  hughes@mathematics.byu.edu, vishnu@neo.phys.wits.ac.za,
  arjunkar@phas.ubc.ca

ABSTRACT: We use deep neural networks to machine learn correlations between knot invariants in various dimensions. The three-dimensional invariant of interest is the Jones polynomial $J(q)$, and the four-dimensional invariants are the Khovanov polynomial $\mathrm{Kh}(q, t)$, smooth slice genus $g$, and Rasmussen's $s$-invariant. We find that a two-layer feed-forward neural network can predict $s$ from $\mathrm{Kh}(q, -q^{-4})$ with greater than 99% accuracy. A theoretical explanation for this performance exists in knot theory via the now disproven knight move conjecture, which is obeyed by all knots in our dataset. More surprisingly, we find similar performance for the prediction of $s$ from $\mathrm{Kh}(q, -q^{-2})$, which suggests a novel relationship between the Khovanov and Lee homology theories of a knot. The network predicts $g$ from $\mathrm{Kh}(q, t)$ with similarly high accuracy, and we discuss the extent to which the machine is learning $s$ as opposed to $g$, since there is a general inequality $|s| \leq 2g$. The Jones polynomial, as a three-dimensional invariant, is not obviously related to $s$ or $g$, but the network achieves greater than 95% accuracy in predicting either from $J(q)$. Moreover, similar accuracy can be achieved by evaluating $J(q)$ at roots of unity. This suggests a relationship with $SU(2)$ Chern–Simons theory, and we review the gauge theory construction of Khovanov homology which may be relevant for explaining the network's performance.

## 1  Introduction

Knots and links[1] are defined most naturally as embeddings of disjoint circles $S^1$ in a three-dimensional ambient space, often taken to be $S^3$ or $\mathbb{R}^3$. However, the numerous isotopy

---

[1]In this work, we will focus mostly on knots, though much of the discussion to follow will apply to links as well.

invariants associated with a knot may be more appropriately defined in two, three, or four dimensions.[2] While a completely dimension independent theory of knots and their invariants is currently out of reach, progress has been made in relating invariants which are naïvely defined in different dimensions.

Perhaps the most famous example of such a relationship involves the Jones polynomial $J(q)$ of a knot, which is a Laurent polynomial in $q$ with integer coefficients. The original definition by Jones [1, 2] is intrinsically two-dimensional, as $J(q)$ is computed by the trace in a Hecke algebra of a braid representative, which is a two-dimensional projection of the knot (a knot diagram) that has been cut open. However, soon after, Witten discovered [3] that the evaluation of $J(q)$ at a specific root of unity, $q = e^{2\pi i/(k+2)}$, is equal to the appropriately normalized expectation value of a fundamental representation Wilson loop operator along the knot in three-dimensional Chern–Simons gauge theory on $S^3$ with gauge group $SU(2)$ and integer coupling constant $k$. This intrinsically three-dimensional perspective on $J(q)$ led to a deeper understanding of various objects in both knot theory and quantum field theory, and in some sense the problem of finding this three-dimensional perspective in the first place was one of the greatest mysteries about the Jones polynomial.[3]

In a later development, Khovanov showed [4] that $J(q)$ may be categorified. What this means is that there exists a $\mathbb{Z} \times \mathbb{Z}$-graded homology theory $\mathcal{H}(L)$, known as Khovanov homology, associated to a link $L$, for which the graded Euler characteristic is $J(q)$. More precisely, there is a finite decomposition

$$\mathcal{H}(L) = \bigoplus_{m,n} \mathcal{H}^{m,n}(L)\,, \tag{1.1}$$

where $\mathcal{H}^{m,n}(L)$ are vector spaces over $\mathbb{Q}$,[4] with $m \in \mathbb{Z}$ and $n \in 2\mathbb{Z} + |L|$, where $|L|$ denotes the number of components in $L$. Then $J(q)$ is given by

$$(q + q^{-1})J(q^2) = \sum_{\substack{m,n \\ \mathcal{H}^{m,n}(L) \neq 0}} \dim \mathcal{H}^{m,n}(L)(-1)^m q^n\,. \tag{1.2}$$

The original definition of $\mathcal{H}(L)$, as explained succinctly in [5], is again intrinsically two-dimensional. Just as with $J(q)$, $\mathcal{H}(L)$ is computed using a two-dimensional projection of the link, and the main technical challenge in [4] is to show that it is actually an invariant (*i.e.*, independent of the chosen projection).

---

[2]Examples of invariants for each of these dimensions are crossing number, hyperbolic volume, and smooth slice genus, respectively. There are also physics inspired invariants which are naturally defined in five or six dimensions, some of which will be important for our discussion.

[3]Indeed, Witten wrote [3] that "the challenge of the knot polynomials has been to... learn what it is that is three dimensional about two dimensional conformal field theory."

[4]Khovanov homology may be defined more generally over other rings and fields. Indeed, Khovanov's original definition was over the ring $\mathbb{Z}[c]$, where $c$ has degree two.

Despite its two-dimensional definition in [4], there are intimations that $\mathcal{H}(L)$ has a higher-dimensional interpretation, just as $J(q)$ does. One of the most direct hints appears in the work of Rasmussen [6], who showed that for a knot $K$ the homology $\mathcal{H}(K)$ contains enough information to extract an even integer $s$ (the aptly named "$s$-invariant" of $K$) whose magnitude supplies a lower bound for the smooth slice genus $g$:

$$|s(K)| \le 2g(K)\,. \tag{1.3}$$

The smooth slice genus of a knot $K$ is the least integer $g$ such that there exists a smoothly embedded orientable surface $\Sigma \subset D^4$ with $K = \partial\Sigma \subset \partial D^4 = S^3$. In this case $\Sigma$ is called a slice surface for the knot $K$, and due to the fact that slice surfaces are embedded in the disk $D^4$, the invariant $g$ is intrinsically four-dimensional. More generally, linear transformations between $\mathcal{H}(L)$ and $\mathcal{H}(L')$ can be associated with link cobordisms: properly embedded orientable surfaces $\Sigma \subset S^3 \times I$ with $\partial\Sigma = L \sqcup L' \subset S^3 \times \partial I$. Upon concatenating cobordisms, the associated linear transformations simply compose. (Rasmussen's arguments make use of the fact that a punctured slice surface for $L$ can be viewed as a cobordism between $L$ and the unknot.) This strongly suggests that $\mathcal{H}(L)$ has another definition which is natural in four or higher dimensions. Such a definition was proposed by Witten [7], who constructed a candidate for $\mathcal{H}(L)$ as the Hilbert space of a certain five-dimensional gauge theory.[5]

While the Jones polynomial is a success story of the general program of freeing invariants from their dimension dependent definitions, the situation with Khovanov homology, the $s$-invariant, and the slice genus remains unclear. In particular, the connection between $\mathcal{H}(L)$ and $s$ described by Rasmussen is not entirely geometric, which makes it difficult to assign a particular dimension to the definition of $s$ even if we assume $\mathcal{H}(L)$ is a natural four- or five-dimensional invariant. It is therefore worthwhile to understand what relationships, if any, exist beyond the obvious ones between $J(q)$, $\mathcal{H}(L)$, $s$, and $g$. In addition to uncovering new structure in physics, finding different perspectives on these objects could also be helpful in addressing important questions in pure mathematics. One such outstanding question is the smooth four-dimensional Poincaré conjecture (SPC4), which asserts that any manifold that is homotopy equivalent to $S^4$ is also diffeomorphic to $S^4$. Because (1.3) informs us that $g$ must be positive whenever $s \ne 0$, [11] develops a strategy for constructing exotic four-spheres. Techniques for producing potential counterexamples to SPC4 are also proposed in [12], which rely on showing that certain topologically slice knots are smoothly slice.[6]

---

[5]Actually, the five-dimensional theory in question is not ultraviolet complete. This fact is discussed at length in [7], and a more powerful formulation in six dimensions is proposed. There are also other physics-based proposals for Khovanov homology [8–10], but we will focus on the approach described in [7].

[6]If $\Sigma$ is a locally flat disk properly embedded in $D^4$ it is called a topologically slice disk, and $K = \partial\Sigma \subset \partial D^4$ is a topologically slice knot. A knot is smoothly slice when its slice genus is zero.

A powerful method for establishing relationships between knot invariants has emerged recently in the form of deep neural networks [13, 14]. The rough idea is to use a dataset to train a universal function approximator to predict one knot invariant from another. Such techniques have been applied fruitfully in knot theory due to the enormous amount of data available on knot invariants, and more general machine learning explorations have yielded new perspectives on the unknotting problem [15] and the volume conjecture [16].[7,8]

The purpose of this paper is to employ these deep neural network techniques to extract subtle correlations between $J(q)$, $\mathcal{H}(L)$, $s$, and $g$, with the broader goal of understanding the properties of each invariant that are surprising from a dimensional perspective. We find strong correlation between properties of $\mathcal{H}(L)$ and $s$, partially explained by the so-called knight move conjecture in knot theory.[9] Another correlation points toward a novel relationship between a specialization of Khovanov homology and the $s$-invariant, similar to (but distinct from) the standard passage through Lee homology [6]. Perhaps more surprisingly, we discover large correlation between $J(q)$ or evaluations at roots of unity of $J(q)$ and the invariants $s$ and $g$. We do not know of a conjecture in knot theory which would explain this, and the physical picture is also unclear. We may phrase this issue as a question, following [3]: what is it that is four-dimensional about the Jones polynomial?[10]

Four sections follow. In Section 2, we review the two-dimensional construction of Khovanov homology and Rasmussen's construction of the $s$-invariant through Lee homology. This will aid our analysis of the mysterious correlation between an unusual specialization of $\mathcal{H}(L)$ and $s$. In Section 3, we briefly review the gauge theoretic ideas of [3, 7] which are relevant for our speculations concerning what higher dimensional information is contained in the Jones polynomial. In Section 4, we recall some standard facts about neural networks and give an overview of our experiments and results in extracting correlations among knot invariants. We conclude in Section 5 with a discussion and speculate on the possible knot and gauge theoretic explanations of our results.

---

[7]One of the main surprises in [14, 16] is that a form of the volume conjecture holds approximately for many knots even for the fundamental representation Jones polynomial. The results of [14, 16] could have been partially anticipated from the analytically continued Chern–Simons theory by physicists familiar with the main ideas of [17]. By contrast, we do not know of a physical explanation for the results we find here, even with the apparently most relevant work [7] in mind. We will comment further on this issue later.

[8]Concurrent with this paper, machine learning methods were applied to motivate a theorem relating the signature, slope, injectivity radius, and volume of hyperbolic knots [18, 19].

[9]Although a counterexample to the knight move conjecture is presented in [20], the statement holds for knots quite broadly and can in fact be proven for certain classes of knots.

[10]Anticipated in [17, 21], discussed in [7], and explained in greater detail in [22], there is a clear sense in which $J(q)$ is the result of a four-dimensional gauge theory calculation. But, as we will discuss, this statement alone does not explain our findings.

## 2 Khovanov homology and the $s$-invariant

### 2.1 Khovanov homology

In what follows, let $D$ be a diagram of the link $L$. Then each crossing of $D$ can be smoothed in one of two different ways, which we call the 0– and 1–smoothings, respectively (see Figure 1). Suppose that $D$ has $n_+$ positive crossings and $n_-$ negative crossings and that we fix an ordering on the combined set of $n = n_+ + n_-$ crossings. Then for for each vertex $v \in \{0,1\}^n$ of the $n$–dimensional cube $[0,1]^n$ we define a total smoothing $D_v$ by applying the smoothing specified by the $k$-th coordinate of $v$ to the $k$-th crossing of $D$. After applying these smoothings the resulting diagram $D_v$ will be a family of disjoint embedded circles in the plane of projection. Define $s(v)$ to be the number of connected components of $D_v$, and let $r(v)$ denote the sum of the components of $v$, *i.e.*, the number of 1–smoothings performed on $D$ to obtain $D_v$. Now let $V$ be a graded $\mathbb{Q}$–vector space with basis $\{\mathbf{v}_+, \mathbf{v}_-\}$, where $\deg \mathbf{v}_+ = 1$ and $\deg \mathbf{v}_- = -1$. For each total smoothing $D_v$ define the vector space $\mathbb{V}_v = V^{\otimes s(v)}\{r(v) + n_+ - 2n_-\}$. Here, we let $W\{k\}$ denote the graded vector space obtained by shifting the gradings of elements of $W$ by $k$. Moreover, to each component of $D_v$ we associate one of the tensor product factors of $\mathbb{V}_v$ via some one-to-one correspondence.

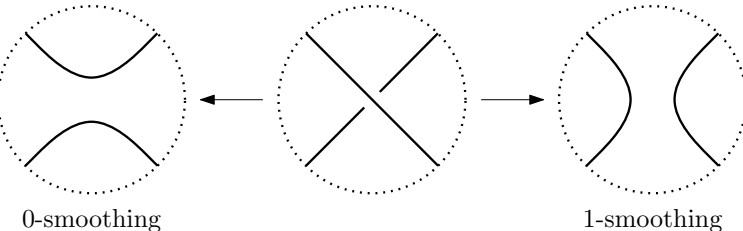

0-smoothing                     1-smoothing

**Figure 1**: Two smoothings of a crossing.

Now, suppose that $v = \{v_1, \ldots, v_n\}$ and $v' = \{v'_1, \ldots, v'_n\}$ are vertices in $\{0,1\}^n$ that agree in all but their $j$-th coordinates, where $v_j = 0$, $v'_j = 1$, and $v_i = v'_i$ for $i \neq j$. It follows then that the sequences of smoothings used to obtain $D_v$ and $D_{v'}$ differ only at the $j$-th crossing. If $e$ is the edge of the cube $[0,1]^n$ connecting $v$ and $v'$, then we associate an edge map $\varphi_e : \mathbb{V}_v \to \mathbb{V}_{v'}$ to $e$ as follows. First, if changing the smoothing performed at the $j$-th crossing from a 0–smoothing to a 1–smoothing results in a pair of circles of $D_v$ merging into a single circle, then on the corresponding tensor product factors of $\mathbb{V}_v$ we define the map $\varphi_e$ as

$$
\begin{aligned}
\mathbf{v}_- \otimes \mathbf{v}_- &\mapsto 0\,, & \mathbf{v}_+ \otimes \mathbf{v}_+ &\mapsto \mathbf{v}_+\,, \\
\mathbf{v}_- \otimes \mathbf{v}_+ &\mapsto \mathbf{v}_-\,, & \mathbf{v}_+ \otimes \mathbf{v}_- &\mapsto \mathbf{v}_-\,.
\end{aligned}
\tag{2.1}
$$

On the other hand, if changing the smoothing performed at the $j$-th crossing from a 0–smoothing to a 1–smoothing corresponds to a single circle of $D_v$ splitting into two circles, then we define $\varphi_e$ on the associated tensor product factors of $\mathbb{V}_v$ as

$$\mathbf{v}_- \mapsto \mathbf{v}_- \otimes \mathbf{v}_- , \qquad \mathbf{v}_+ \mapsto \mathbf{v}_+ \otimes \mathbf{v}_- + \mathbf{v}_- \otimes \mathbf{v}_+ . \tag{2.2}$$

The map $\varphi_e$ is then defined to be the identity map on the unaffected tensor product factors of $\mathbb{V}_v$. Defined in this way, it is not difficult to show that the maps $\varphi_e$ commute around any 2–dimensional face of the cube $[0,1]^n$.

We can now define the Khovanov chain complex $\mathcal{C}(D) = (C^k(D), d^k)$ associated to the diagram $D$, with chain groups

$$C^k(D) = \bigoplus_{r(v)=k+n_-} \mathbb{V}_v , \tag{2.3}$$

and differential $d^k : C^k(D) \to C^{k+1}(D)$ given by

$$d^k = \sum \epsilon_e \varphi_e . \tag{2.4}$$

The sum in (2.4) is taken over all edges joining pairs of vertices $v$ and $v'$, with $r(v) = k + n_-$ and $r(v') = r(v) + 1$, and the signs $\epsilon_e \in \{-1, 1\}$ are chosen so that the squares of the resulting differentials are zero (such a choice of the $\epsilon_e$ is always possible). Elements of $C^k(D)$ inherit a grading from the vertex vector spaces $\mathbb{V}_v$, which we refer to as the quantum grading, and we define the homological grading of an element of $C^k(D)$ to be $k$. It can then be checked that the differentials $d^k$ preserve the quantum grading of elements of $C^k(D)$, but increase their homological grading by 1. The homology $\mathcal{H}(L)$ of the resulting chain complex $\mathcal{C}(D) = (C^k(D), d^k)$ is therefore bigraded, with gradings induced by the homological and quantum gradings on $\mathcal{C}(D)$. This homology $\mathcal{H}(L)$, called the Khovanov homology of $L$, was first defined in [4], where Khovanov showed that up to bigrading-preserving isomorphism it only depends on the underlying isotopy class of the link $L$. For a knot $K$ the homological grading of $\mathcal{H}(K)$ takes values in $\mathbb{Z}$, while the quantum grading values are all odd integers. It is often convenient to express the Khovanov homology of a knot or link in terms of its Poincaré series, which is a Laurent polynomial defined by

$$\mathrm{Kh}(L; q, t) = \sum_{i,j} \dim(\mathcal{H}^{i,j}(L)) t^i q^j , \tag{2.5}$$

where $\mathcal{H}^{i,j}(L)$ denotes the subspace of $\mathcal{H}(L)$ consisting of all elements with homological grading equal to $i$ and quantum grading equal to $j$. We refer to $\mathrm{Kh}(L; q, t)$ as the Khovanov polynomial of $L$.

## 2.2 Lee homology and Rasmussen's $s$-invariant

In [23], Lee defined a modification of this homology theory, by constructing a chain complex with the same chain groups $C^k(D)$ as Khovanov's original theory, but with modified edge maps $\varphi'_e : \mathbb{V}_v \to \mathbb{V}_{v'}$. When the edge $e$ corresponds to the merging of two circles the map $\varphi'_e$ is defined on the affected tensor product factors as

$$
\begin{aligned}
\mathbf{v}_- \otimes \mathbf{v}_- &\mapsto \mathbf{v}_+ \,, & \mathbf{v}_+ \otimes \mathbf{v}_+ &\mapsto \mathbf{v}_+ \,, \\
\mathbf{v}_- \otimes \mathbf{v}_+ &\mapsto \mathbf{v}_- \,, & \mathbf{v}_+ \otimes \mathbf{v}_- &\mapsto \mathbf{v}_- \,.
\end{aligned}
\tag{2.6}
$$

When $e$ corresponds to a circle being split, the edge map $\varphi'_e$ is defined by

$$
\mathbf{v}_- \mapsto \mathbf{v}_- \otimes \mathbf{v}_- + \mathbf{v}_+ \otimes \mathbf{v}_+ \,, \qquad \mathbf{v}_+ \mapsto \mathbf{v}_+ \otimes \mathbf{v}_- + \mathbf{v}_- \otimes \mathbf{v}_+ \,.
\tag{2.7}
$$

Notice now that the resulting differential $d^k_{\text{Lee}} : C^k(D) \to C^{k+1}(D)$ is no longer grading-preserving, but is still non-decreasing in the quantum grading. As a result of this, we can define a filtration on $C^k(D)$, which is a nested sequence of subspaces

$$
\{0\} = F^n C^k(D) \subseteq F^{n-1} C^k(D) \subseteq \ldots \subseteq F^m C^k(D) = C^k(D) \,,
\tag{2.8}
$$

where $F^j C^k(D)$ is defined to be the space of all elements of $C^k(D)$ with quantum grading greater than or equal to $j$. Note then that $d^k_{\text{Lee}}(F^j C^k(D)) \subseteq F^j C^{k+1}(D)$ for all $j$ and $k$, and hence we obtain a filtered chain complex $\mathcal{C}_{\text{Lee}}(D) = (C^k(D), d^k_{\text{Lee}})$.

The filtration on $\mathcal{C}_{\text{Lee}}(D)$ gives rise to a spectral sequence, whose $E^1$ page is the ordinary Khovanov homology $\mathcal{H}(L)$, and whose $E^\infty$ page gives the homology $\mathcal{H}_{\text{Lee}}(L)$ of the chain complex $\mathcal{C}_{\text{Lee}}(D)$. Lee [23] showed that for a knot $K$, the homology $\mathcal{H}_{\text{Lee}}(K)$ is isomorphic to $\mathbb{Q} \oplus \mathbb{Q}$. Rasmussen [6] then proved that these $\mathbb{Q}$–summands both have homological grading zero, and have quantum gradings $s \pm 1$ for some $s \in 2\mathbb{Z}$. This $s$ is an invariant of the knot $K$, called the Rasmussen $s$-invariant.

The Lee spectral sequence gives rise to the following important corollary: for any knot $K$, the Khovanov polynomial can be factored as

$$
\text{Kh}(K; q, t) = q^s(q + q^{-1}) + \sum_{\ell \geq 1} f_{2\ell}(q, t)(1 + tq^{4\ell}) \,,
\tag{2.9}
$$

for some Laurent polynomials $f_{2\ell} \in \mathbb{N}[q^{\pm 1}, t^{\pm 1}]$ with positive coefficients, where $s$ is the Rasmussen $s$-invariant of the knot. The so-called knight move conjecture posits that $f_{2\ell}(q, t) = 0$ for all $\ell > 1$. Although the conjecture is known to be false — Manolescu and Marengon [20] present a counterexample by way of a diagram with 38 crossings — it is known to hold for

alternating [24] and quasi-alternating [25] knots, knots with unknotting number less than or equal to two [26], and homologically thin knots [6].[11]

In the case of homologically thin knots (2.9) implies that the Khovanov homology is entirely determined by the Jones polynomial and the $s$-invariant. Lee proved that alternating knots are homologically thin, and Rasmussen showed that for alternating knots the $s$-invariant is equal to the classical signature $\sigma$. Hence the Khovanov homology of alternating knots is entirely determined by the signature $\sigma$ and Jones polynomial $J(q)$.

## 2.3 Rasmussen's $s$-invariant and the slice genus

In addition to proving that $s$ is a knot invariant, Rasmussen also showed that it provides the lower bound on the slice genus of a knot given in (1.3). To see why this is, let $\Sigma \subset S^3 \times I$ be a smoothly embedded surface with $\partial \Sigma = L \sqcup L'$, where $L \subset S^3 \times \{0\}$ and $L' \subset S^3 \times \{1\}$ are both links. After a small perturbation of $\Sigma$, we may assume that the projection map $h : S^3 \times I \to I$ restricts to a Morse function $h|_\Sigma : \Sigma \to I$. Let $L_t := h|_\Sigma^{-1}(t) = \Sigma \cap (S^3 \times \{t\})$ for all $t \in I$. Then $L_t$ will be a non-singular link embedded in $S^3 \times \{t\}$ for all but finitely many $t \in I$, which correspond to the critical points of the Morse function $h|_\Sigma$. Furthermore, if we select a projection map $\pi : S^3 \to S^2$, then $\pi$ induces a projection map $\pi_t : S^3 \times \{t\} \to S^2$ for all $t \in I$. After another small perturbation we may assume that $D_t := \pi_t(L_t)$ is a regular link diagram of $L_t$ in $S^2$ for all but finitely many points $0 < t_1 < t_2 < \cdots < t_m < 1$.

Starting at $t = 0$ and moving up, we see that between any two values $t_i$ and $t_{i+1}$ the diagram $D_t$ changes only by isotopy in $S^2$. When passing any $t_i$ value, the diagram $D_t$ changes in one of the following ways:

1. By applying a single Reidemeister move to the diagram $D_t$.

2. By adding a small circle that is disjoint from the rest of the diagram (corresponding to a local minimum of the Morse function $h|_\Sigma$).

3. By removing a small circle that is disjoint from the rest of the diagram (corresponding to a local maximum of the Morse function $h|_\Sigma$).

4. By performing a saddle surgery to to the diagram (corresponding to a saddle point of the Morse function $h|_\Sigma$).

---

[11]Homologically thin knots are knots $K$ for which all nontrivial Khovanov groups $\mathcal{H}^{i,j}(K)$ satisfy $j - 2i \in \{n - 1, n + 1\}$ for some integer $n$. In fact, for such knots $j - 2i = s \pm 1$, where $s$ is the Rasmussen $s$-invariant.

Then by setting $s_0 = 0$, $s_m = 1$, and selecting points $s_1, \ldots, s_{m-1}$ which satisfy $t_k < s_k < t_{k+1}$, we may cut $\Sigma$ at each of the heights $s_k$, thereby splitting $\Sigma$ into a union of elementary cobordisms $\Sigma = \Sigma_1 \cup \Sigma_2 \cup \cdots \cup \Sigma_m$, where each $\Sigma_i$ contains precisely one of the singular points $t_i$. Note that for each $k$ the boundary of $\Sigma_k$ splits as $\partial \Sigma_k = L_{s_{k-1}} \sqcup L_{s_k}$, and to each $\Sigma_k$ Rasmussen associates a map $\phi_k : \mathcal{H}_{\mathrm{Lee}}(L_{s_{k-1}}) \to \mathcal{H}_{\mathrm{Lee}}(L_{s_k})$ on the Lee homology of its boundary components. Each of these maps is an isomorphism, and to the full cobordism $\Sigma$ Rasmussen defines an isomorphism $\phi_\Sigma : \mathcal{H}_{\mathrm{Lee}}(L) \to \mathcal{H}_{\mathrm{Lee}}(L')$ given by the composition $\phi_\Sigma = \phi_m \circ \cdots \circ \phi_2 \circ \phi_1$.

Although each $\phi_k$ is an isomorphism, they behave differently with respect to the induced filtration on their respective Lee homology. When the singular level $t_k$ corresponds to performing a single Reidemeister move on the diagram $D_{s_{k-1}}$, the map $\phi_k$ is a filtered map of degree zero. On the other hand, when $t_k$ corresponds to adding or removing a small circle to the diagram $D_{s_{k-1}}$, the map $\phi_k$ is a filtered map of degree one. Finally, $\phi_k$ will be filtered of degree $-1$ whenever $t_k$ corresponds to performing a saddle surgery on the diagram. It follows then that the map $\phi_\Sigma$ will be filtered of degree $v - e = \chi(\Sigma)$, where $v$ is the number of local maximum and minimum points in $\Sigma$, $e$ is the number of saddle points, and $\chi(\Sigma)$ is the Euler characteristic of $\Sigma$.

Suppose now that $K \subset S^3$ is a knot, and that $F$ is a slice surface for $K$ of genus $g_F$. In other words, $F$ is a compact oriented surface, smoothly embedded in $D^4$, with $\partial F = K$. By removing a disk from $F$, we obtain a cobordism $\Sigma$ between $K$ and the unknot, which induces a filtered map $\phi_\Sigma$ of degree $\chi(\Sigma) = \chi(F) - 1 = -2g_F$. However, as $\mathcal{H}_{\mathrm{Lee}}(K) \cong \mathbb{Q} \oplus \mathbb{Q}$ is supported at quantum gradings $s \pm 1$, while $\mathcal{H}_{\mathrm{Lee}}(\bigcirc) \cong \mathbb{Q} \oplus \mathbb{Q}$ is supported at quantum gradings $\pm 1$, it follows that $\mathrm{degree}(\phi_\Sigma) = -2g_F \leq -s$. By repeating the same calculation with the mirror image of $K$ (and observing that taking them mirror image of a knot changes the $s$-invariant by a sign), we obtain the bound in (1.3).

## 3   Gauge theory and knot invariants

There are well-known gauge theory constructions for some of the knot invariants under consideration, and it will be useful for us to review them. To begin, we discuss the three-dimensional Chern–Simons [3] and four-dimensional super-Yang–Mills [7, 21, 22] pictures of $J(q)$. We also briefly review the five- and six-dimensional pictures of $\mathcal{H}(L)$ from [7].

### 3.1  3d Chern–Simons theory

Let us recall how the Jones polynomial arises in Chern–Simons theory [3]. The Chern–Simons action is

$$S_{\text{CS}}(A) = \frac{k}{4\pi} \int_M \text{tr}\big(A \wedge dA + \frac{2}{3} A \wedge A \wedge A\big), \tag{3.1}$$

where $M$ is a three manifold and the one-form $A$ is a $\mathfrak{g}$ valued connection, with $\mathfrak{g}$ the Lie algebra of the gauge group $G$. The Chern–Simons level $k$ is integer quantized to ensure single valuedness of $e^{iS_{\text{CS}}}$ under large gauge transformations. Chern–Simons theory is a topological quantum field theory, meaning that correlation functions are independent of the metric on $M$. The operators of interest are Wilson loops:

$$U_R(\gamma) = \text{tr}_R \, \mathcal{P} \exp\left(\oint_\gamma A\right), \tag{3.2}$$

where $R$ identifies a representation of the gauge group $G$ and $\mathcal{P}$ denotes path ordering. Taking $M$ to be $S^3$, $\gamma$ to be a knot $K$ embedded as $S^1 \subset S^3$, and $G = SU(2)$, the colored Jones polynomial is the unknot normalized vacuum expectation value of the Wilson loop operator:

$$J_n(K; q = e^{2\pi i/(k+2)}) = \frac{\langle U_n(K)\rangle}{\langle U_n(\bigcirc)\rangle}, \tag{3.3}$$

where

$$\langle U_n(K)\rangle = \frac{1}{Z} \int_{\mathcal{U}} [DA] \, U_n(K) \, e^{iS_{\text{CS}}(A)}, \qquad Z = \int_{\mathcal{U}} [DA] \, e^{iS_{\text{CS}}(A)}, \tag{3.4}$$

and $n$ labels the $n$-dimensional irreducible representation of $SU(2)$. The path integrals are taken over the space $\mathcal{U}$ of $\mathfrak{su}(2)$ connections modulo gauge transformations. The shift $k \mapsto k+2$ in the definition of $q$ in (3.3) accounts for one loop corrections to the path integral. (The 2 is the dual Coxeter number of $SU(2)$.) The Jones polynomial $J(q) = J_2(K; q)$ corresponds to working in the fundamental representation of $SU(2)$.

The Chern–Simons path integral gives a manifestly three-dimensional definition for evaluations of $J(q)$. However, it does not explain at least two very basic facts about $J(q)$ which are obvious from the original two-dimensional definition. The path integral does not explain why the Jones polynomial should be a polynomial in the first place, and also why it should have integer coefficients. The second of these is perhaps the deeper issue, and it is addressed by Khovanov homology. In Khovanov homology, the coefficients of $J(q)$ are interpreted as (graded) dimensions of homology groups, as in (1.1). However, to understand the appearance of Khovanov homology in gauge theory, following [7], we must venture beyond Chern–Simons theory.[12]

---

[12]It is not actually proven in [7] that the construction described there coincides with Khovanov homology. This point will not concern us in this work. We will continue to describe the homology theory constructed

### 3.2 4d $\mathcal{N} = 4$ super-Yang–Mills

The first step in constructing Khovanov homology from gauge theory is rewriting the Chern–Simons path integral on $M$ as a four-dimensional path integral in (a twisted version of) $\mathcal{N} = 4$ super-Yang–Mills theory on a manifold with boundary $V = M \times \mathbb{R}_+$. This construction originated in a study of integration cycles in quantum mechanics and Chern–Simons theory [17, 21], as the equivalence can sometimes necessitate considering Chern–Simons theory with a rather exotic integration cycle. As the two path integrals are equal, the $\mathcal{N} = 4$ path integral will produce the Jones polynomial $J(q)$ [22].

The details of the construction will not be so important for us, but it will be useful to understand both what the computation entails as well as just what the variable $q$ represents in the four-dimensional language. In a certain sense, this will tell us something about the Jones polynomial which is intrinsically four-dimensional. In the $\mathcal{N} = 4$ $SO(3)$ gauge theory, there is a theta angle[13] multiplying the following term in the action:

$$S_{\mathcal{N}=4} \supset \frac{\theta^\vee}{64 h^\vee \pi^2} \int_V \epsilon^{\mu\nu\alpha\beta} \operatorname{Tr} F_{\mu\nu} F_{\alpha\beta} , \tag{3.5}$$

where the trace is taken in the adjoint representation and $h^\vee = 1$ is the dual Coxeter number of $SO(3)$. This term is topological in nature, and counts the "instanton number" of a classical solution to the $\mathcal{N} = 4$ field equations. If the instanton number of a solution is $n$, the saddle point contribution to the $\mathcal{N} = 4$ path integral from (3.5) is $e^{-in\theta^\vee}$.[14] The Jones polynomial is supposed to be computed by summing over certain supersymmetric field configurations in the twisted $\mathcal{N} = 4$ theory, and this accounts for the full path integral (up to a ratio of one-loop determinants) due to a supersymmetric localization. The ratio of determinants has unit magnitude by supersymmetry, and its sign is determined by the fermion component. As these sums of exponentials are supposed to reproduce the Jones polynomial, the variable $q$ must be simply the weight of the topological term in the path integral:

$$q = \exp(-i\theta^\vee) . \tag{3.6}$$

---

in [7] as Khovanov homology, and it is enough for us that [7] constructs some kind of enhanced Floer-like homology theory which has the Jones polynomial as its graded Euler characteristic.

[13]Following [7], we denote this angle by $\theta^\vee$, as it appears in a gauge theory with gauge group $G^\vee$ that is Langlands or GNO dual to the original Chern–Simons gauge group $G$. For the Jones polynomial, which arises for Chern–Simons gauge group $SU(2)$, the dual group is $G^\vee = SO(3)$, though in this case the Lie algebras coincide.

[14]We are pretending that the result of the instanton number integral is an integer, but this may not be strictly true. It depends on the boundary conditions on $M \times \mathbb{R}_+$, among other subtle points such as a framing of the spin bundle; see Section 3.5 of [7] for a detailed discussion. The gauge group itself can also play a role, and the fact that $SO(3)$ is not simply connected means that we at least have $n \in \mathbb{Z}/4 + \delta$ where $\delta$ depends on the boundary conditions.

This gives us some geometric intuition for how evaluations of the Jones polynomial can carry four-dimensional information. The quantity $J(e^{-i\theta^\vee})$ is the result of the $\mathcal{N} = 4$ path integral. This still has not told us how to produce Khovanov homology, however. For that, we need yet another dimension.

### 3.3 5d maximal super-Yang–Mills

To get Khovanov homology, we must consider a five-dimensional theory. This is motivated by an idea which is suggested by the form of (1.1): the Jones polynomial should be produced by the graded trace in some vector space. Such graded trace interpretations of path integrals arise when there is an $S^1$ factor of the manifold and the Hilbert space of the Cauchy surface is the vector space in question. As the Jones polynomial was obtained previously as a sum over solutions of four-dimensional supersymmetric equations which are essentially the time-independent versions of the lifted five-dimensional equations, we interpret these solutions as approximate supersymmetric ground states of the five-dimensional theory.

The theory in question ought to reduce at low energies to the twisted $\mathcal{N} = 4$ theory we had previously, which means we are considering 5d maximally supersymmetric Yang–Mills theory on $S^1 \times M \times \mathbb{R}_+$. Due to various issues discussed in [7], taking $M = \mathbb{R}^3$ is the appropriate choice to produce something like Khovanov homology. The topological supercharge $Q$ from the $\mathcal{N} = 4$ theory remains a symmetry, and has an action on the Hilbert space of the five-dimensional theory. The candidate for $\mathcal{H}(K)$ in this construction is the cohomology of this supercharge $Q$ acting on the Hilbert space of the five-dimensional theory on $M \times \mathbb{R}_+$.

The resulting $\mathcal{H}(K)$ is $\mathbb{Z} \times \mathbb{Z}$-graded, as expected of Khovanov homology. The "quantum grading" arises from the instanton number which we discussed in the $\mathcal{N} = 4$ context, and the "homological grading" is due to a fermion number symmetry.[15] Of course, in the five-dimensional formulation, the object we call instanton number is really an operator acting on the Hilbert space of the theory, whereas in the previous $\mathcal{N} = 4$ description it was a term in the action.

We mentioned that the Jones polynomial is produced by a graded trace in the space of approximate supersymmetric ground states. It is also given by the graded trace in the space of exact ground states, as we have just said this is to be identified with $\mathcal{H}(L)$. These computations are equal for the following reason. The space of exact states is constructed in the usual manner of Morse homology. The unbroken supercharge $Q$ acts as a differential on

---

[15]Technically, the homological grading is associated with a certain $R$-symmetry generator. See [7] for a discussion of why this object is essentially a fermion number operator.

the chain complex of critical points (approximate ground states), and the cohomology of $Q$ yields the true space of ground states. In physics language, this incorporates the effects of instanton corrections to the energy levels of the approximate ground states. However, due to supersymmetry, states can only be lifted from zero energy in boson-fermion pairs with the same instanton number, which leaves the graded trace (a supersymmetric equivariant index) unchanged.

## 3.4  6d (0,2) theory

Though it is not crucial for our discussion, we note that an ultraviolet complete setup is provided in [7] in the form of dimensional reduction from the (appropriately twisted) $(0, 2)$ superconformal field theory in six dimensions.[16] This theory has no Lagrangian description, and is known largely as the worldvolume theory of M5-branes in M-theory. Compactifying the theory on a surface leads to the celebrated AGT correspondence between Liouville theory on the surface and four-dimensional $\mathcal{N} = 2$ superconformal field theories [29].

To get Khovanov homology in this theory, we consider a six-manifold $\mathbb{R}_t \times M \times \mathbb{R}^2$. The factor $\mathbb{R}^2$ admits a $U(1)$ action which has a fixed point at the origin, which we call $p \in \mathbb{R}^2$. Then the proposal made in [7] is that $\mathcal{H}(K)$ is the cohomology of an unbroken supercharge $Q$ acting on the Hilbert space of this theory when the Cauchy surface is $M \times \mathbb{R}^2$, and to obtain the result for a nontrivial knot we need a time-independent surface operator inserted on $\mathbb{R}_t \times K \subset \mathbb{R}_t \times M \times p$. The $U(1)$ action is generated by an object $P$ which obeys $[P, Q] = 0$ and reduces to the instanton number in the previously discussed five-dimensional theory.

## 3.5  Rasmussen's $s$-invariant and gauge theory

We have discussed the Jones polynomial and Khovanov homology from the perspective of gauge theory following [7] in the interest of understanding what four-dimensional information the Jones polynomial may contain. Previously, we have seen that the $s$-invariant can often be extracted from Khovanov homology along with the Jones polynomial, so there is at least some naïve relationship between them. We now briefly describe two attempts to construct the $s$-invariant in gauge theory.

The first attempt to construct $s$ in gauge theory uses instanton Floer homology. The resulting knot invariant, $s^\sharp$, was defined by Kronheimer and Mrowka [30] and turns out not to be equal to $s$, but the two do share some qualitative properties. Instanton Floer homology

---

[16]See also [27, 28] for the connection between the five-dimensional super-Yang–Mills theory and the six-dimensional $(0, 2)$ theory compactified on $S^1$.

is related to Khovanov homology, but the two are not exactly the same. Roughly speaking, instanton Floer homology is constructed as the Morse homology of the space of gauge fields on a three-manifold $M$ with the Chern–Simons action playing the role of the Morse function.[17] Critical points are flat connections, and the gradient flow equations are the self-dual instanton equations $F^+ = 0$ of four-dimensional Yang–Mills theory on $M \times \mathbb{R}$. The invariant $s^\sharp$ is then computed from instanton Floer homology in a manner very reminiscent of how $s$ arises from Khovanov homology, as we reviewed in Section 2. One constructs a cobordism between $K$ and the unknot, and subsequently studies the properties of the map between instanton Floer homologies induced by this cobordism. It has been shown that $s^\sharp \neq s$ even for the trefoil [30], but the precise relationship between the two quantities is unclear as their magnitudes are both lower bounds on twice the slice genus. One fact which seems worth mentioning is that the instanton Floer homology used in [30] is naturally $\mathbb{Z}/4\mathbb{Z}$-graded,[18] while Khovanov homology is fully $\mathbb{Z}$-graded (arising from the fermion number in 5d super-Yang–Mills). Perhaps more insight could be gained in this direction by considering instanton Floer homology with gauge group $SU(N)$ and using level-rank duality [31].

The second attempt at constructing an invariant like $s$ from gauge theory makes use of the techniques in [7] which we reviewed above. There was a conjecture in knot theory which we discussed around (2.9), now known to be false in general [20], involving the Khovanov polynomial $\mathrm{Kh}(K; q, t)$ of a knot and Rasmussen's $s$-invariant [11].[19] It can be proven for homologically thin knots and states

$$q^s = \frac{\mathrm{Kh}(q, -q^{-4})}{q + q^{-1}} = \frac{\mathrm{Kh}(K; q, -q^{-4})}{\mathrm{Kh}(\bigcirc; q, -q^{-4})} \, . \tag{3.7}$$

Indeed, the Khovanov polynomial for the unknot is independent of the variable $t$. The expression (3.7) is structurally similar to (3.3). In light of the observation that Khovanov homology may have a formulation as the ground state Hilbert space (on $\mathbb{R}^3 \times \mathbb{R}_+$ with suitable operator insertions) of a five-dimensional gauge theory, this conjecture immediately suggests a trace formula for $s$, at least when $K$ is thin:

$$\lim_{k \to \infty} k \log \left( \frac{Z_5[K; k]}{Z_5[\bigcirc; k]} \right) = 2\pi i s \, , \tag{3.8}$$

---

[17]This is certainly related to the constructions in [7], and the Morse theory perspective on Khovanov homology is explained in Section 5.3.2 of [7].

[18]As reviewed in Section 5.3.2 of [7], due to the fact that the Chern–Simons action is only well-defined on the space of gauge fields modulo gauge transformations up to multiples of $2\pi k$, standard Floer theory is actually $\mathbb{Z}/4h\mathbb{Z}$-graded, where $h$ is the dual Coxeter number of the gauge group. In [30], the gauge group is $SO(3)$, which has $h = 1$.

[19]To streamline notation, where the meaning is unambiguous, we drop the label $K$ in writing knot polynomial invariants.

where $Z_5[K; k]$ is shorthand for the doubly-graded trace in the cohomology of $Q$ in the Hilbert space on $\mathbb{R}^3 \times \mathbb{R}_+$ with appropriate operator insertions to produce the Khovanov polynomial of the knot $K$ evaluated at quantum grading $q = e^{2\pi i/(k+2)}$ and homological grading $t = -e^{-8\pi i/(k+2)}$. An alternate expression for (3.8), which physicists would call a "saddle point approximation," is[20]

$$Z_5[k] \underset{k \to \infty}{\sim} e^{2\pi is/k + \log 2}. \tag{3.9}$$

This formula is reminiscent of the volume conjecture [8, 32–34] which expresses the volume of the knot complement of a hyperbolic knot as an evaluation of the colored Jones polynomial $J_n(q)$ in the large-$n$ limit. However, unlike the volume conjecture, (3.8) involves only the $n = 2$ fundamental representation of $SU(2)$ in the associated Chern–Simons theory.

The ordinary Jones polynomial is obtained from evaluating the Khovanov polynomial at the specific value $t = -1$, as in (1.1), and this is the point reached above as $k \to \infty$. Moreover, we have $q \to 1$ when $k \to \infty$. We see from this that, at least for certain knots, the analytic structure of the Khovanov polynomial $\mathrm{Kh}(q, t)$ contains information about $s$ near a point where it coincides with the Jones polynomial evaluation, namely $J(1)$. Of course, the function $J(q)$ only appears from $\mathrm{Kh}(q, t)$ on the hypersurface $t = -1$, so we apparently must move off the $t = -1$ surface at a particular angle set by $t = -q^{-4}$ to learn about $s$.

In general, the knight move conjecture can fail, and counterexamples are known in the literature. These counterexamples have higher order terms ($f_{2\ell}(q, t) \neq 0$ for some $\ell > 1$) in the factorization (2.9), which spoils the simple extraction of $s$ from gauge theory that we outlined above. From the physics perspective, the failure of the knight move conjecture corresponds to pairs of additional supersymmetric solutions in the path integral which have quantum numbers that differ by more than one in fermion number or more than four in instanton number.

## 4  Experiments

Machine learning, and in particular neural networks, have been employed in knot theory to predict knot invariants like the slice genus and Ozsváth–Szabó $\tau$-invariant [13] and the

---

[20]We have written this formula as if the graded trace in the Hilbert space of ground states can be also written as a path integral. For reasons described in [7], this is not exactly true. It is true for the Jones polynomial because that is a type of index of an operator, but the most general path integral on $S^1$ will receive contributions from states of nonzero energy. Nevertheless, due to supersymmetric localization, there is a classical solution of the supersymmetric equations with the appropriate quantum numbers; what we cannot say is whether this solution is unique or if there are others which complicate the limit.

hyperbolic volume [14, 16], as well as to solve problems like the unknot recognition problem [15].[21] Neural networks are particularly useful when working with large datasets, and millions of knots have been tabulated together with efficient algorithms for computing their invariants. Such networks are therefore appropriate tools to study the relationships between knot invariants.

A neural network is a function, $f_\theta(\mathbf{v})$, which approximates the relationship between input and output features, $\mathcal{A} : \mathbf{v}_{\text{in}} \to \mathbf{v}_{\text{out}}$, by adjusting its parameters $\theta$ to recreate the assignment $\mathcal{A}$ as closely as possible. A single layer, finite width neural network can approximate any well-behaved function on a compact subset of $\mathbb{R}^N$ [37, 38]. What is noteworthy about the neural networks we employ is that they have relatively small architectures and achieve high accuracy when trained on small fractions of the dataset. This suggests the underlying mathematical structures allow a simple analytic description. In this way, machine learning can function as a discovery tool in mathematics and theoretical physics.[22]

A neural network achieves a non-linear best fit by tuning the elements of weight matrices and the components of bias vectors corresponding to each of $n$ hidden layers — these are collectively termed the hyperparameters or weights $\theta$ — to extremize a loss function. We first prepare the input as a vector $\mathbf{v}_{\text{in}} \in \mathbb{R}^{d_0}$. In the first hidden layer, we obtain a new vector $\mathbf{v}_1 \in \mathbb{R}^{d_1}$:

$$\mathbf{v}_1 = W_\theta^1 \cdot \mathbf{v}_{\text{in}} + \mathbf{b}_\theta^1 , \qquad \mathbf{b}_\theta^1 \in \mathbb{R}^{d_1} . \tag{4.1}$$

As $W_\theta^1$ is a $d_1 \times d_0$ matrix, there are $d_1$ neurons in the first hidden layer. Next, we apply a non-linearity to obtain a second vector, the activation, defined as $\mathbf{a}_1 = \sigma(\mathbf{v}_1) \in \mathbb{R}^{d_1}$. This non-linearity is understood to act elementwise on each component of $\mathbf{v}_1$. For this purpose, we use the Rectified Linear Unit (ReLU), which is written in terms of the Heaviside step function as $\sigma(x) = x\Theta(x)$. This process iterates through $n$ hidden layers specified by new weight matrices $W_\theta^m$ and bias vectors $\mathbf{b}_\theta^m$, $m = 1, \ldots, n$, so that in the end, we have

$$f_\theta(\mathbf{v}_{\text{in}}) = \mathbf{a}_n = \sigma(\mathbf{v}_n) , \qquad \mathbf{v}_n = W_\theta^n \cdot \mathbf{a}_{n-1} + \boldsymbol{b}_\theta^n \in \mathbb{R}^{d_n} . \tag{4.2}$$

We then compare $f_\theta(\mathbf{v}_{\text{in}})$ to the true answer. The batch size determines how many input vectors are evaluated before updating the hyperparameters of the neural network based on extremizing a cross entropy loss function appropriate to a classification problem. The back-propagation algorithm adjusts the hyperparameters layer by layer by computing the gradient of the loss with respect to the weights, iterating backwards from the final layer. One epoch is

---

[21]As well, [35, 36] study many of the same invariants we do from dimensionality reduction and topological data analysis perspectives.

[22]For two success stories in employing machine learning techniques to discover analytic results in mathematical physics, see [16, 39].

a full pass through the set of vectors used in training, and we repeat this process through several epochs. Having fixed the hyperparameters in this manner, the trained neural network's performance is deduced from evaluating $f_\theta$ on a test set complementary to the vectors used for training.

In this work, we typically employ a neural network with two hidden layers, each consisting of 100 nodes.[23] The networks are trained for between 50 and 100 epochs. The training fraction ranges from 10% up to 50%. The machine learned results were stable under adjustments to the neural network architecture. This includes adding more layers, including more neurons per layer, including dropout layers, or using different activation functions.

## 4.1 Data generation

Data for our experiments were generated via two independent methods: random braid words (with special care given to tracking the slice genus) and random knots via the petaluma model [40]. Invariants for these knots were then computed via the KnotTheory package [41, 42] in Mathematica and and Schütz's KnotJob software [43].

By a theorem of Alexander, any knot can be represented as the closure of a braid. Braids in turn may be represented as finite words in the letters $\sigma_1^{\pm 1}, \sigma_2^{\pm 1}, \ldots, \sigma_{n-1}^{\pm 1}$, where $n$ is the number of strands in the braid. Here, $\sigma_i$ denotes the simple braid formed by taking $n$ parallel strands, and adding a single positive crossing between the $i$-th and $(i+1)$-th strands (to obtain $\sigma_i^{-1}$ we add a single negative crossing instead). By forming a random sequence in the letters $\sigma_1^{\pm 1}, \sigma_2^{\pm 1}, \ldots, \sigma_{n-1}^{\pm 1}$ and taking the closure, we obtain a random link.

Our algorithm for generating random braid words required more care, however, to ensure that we could compute the slice genus of a suitable number of the resulting knots. To do this each random braid began with a randomly selected seed braid. These seed braids were of two types: braids from the KnotInfo database [44] for which the slice genus of the closure was known, and randomly generated quasipositive and quasinegative braids.[24]

Given such a seed braid word $\beta$ whose closure has known slice genus, we then randomly inserted braid words of the form $\alpha \sigma_i^{\pm 1} \alpha^{-1}$ into $\beta$. Each such word we inserted into $\beta$ either changed the slice Euler characteristic of the closure by $\pm 1$ or 0, depending on the strands

---

[23]These were constructed with python using TensorFlow with a Keras wrapper. We use Adam for optimization. The essential code is quoted in Appendix A.

[24]An $n$–strand braid is quasipositive if it can be represented as a product of braid words of the form $\alpha \sigma_i \alpha^{-1}$, for some $n$–strand braid $\alpha$ (quasinegative braids are defined similarly). The slice genus of the closure of any quasipositive or quasinegative braid can be easily computed via the slice-Bennequin inequality [45].

involved. Together with the slice genus of the seed braid $\beta$, the number of braid words inserted into $\beta$ allowed us to obtain upper and lower bounds on the slice genus of the resulting knot. These bounds were then combined with slice genus bounds obtained from invariants computed using the KnotTheory and KnotJob software packages, allowing us to exactly pinpoint the slice genus of many of the knots we produced. Finally, we then applied a sequence of random Markov moves to the braids we obtain to further randomize the dataset.

A smaller, independent dataset was also generated using random petal permutations. Indeed, Adams et al. [46] proved that every knot can be represented via a petal diagram, and that such diagrams can be represented by a permutation of the integers $\{1, 2, \ldots, 2n + 1\}$ for some $n$. By sampling random permutations we thus obtained an independent set of random knots, which we again computed invariants for using the KnotTheory and KnotJob software packages. In the case of knots obtained via random petal permutations, we have less information regarding the slice genus, however, and hence these knots were excluded from experiments involving the slice genus.

## 4.2  What does the data look like?

Our dataset includes the Khovanov polynomials, Jones polynomials, Rasmussen $s$-invariants, and signatures of 535239 knots. We also know the slice genus of 82% of these knots (438295 knots). The relationship in (1.3) is known to be saturated for 414615 of the knots. The relationship

$$s(K) = \sigma(K)\,, \tag{4.3}$$

where $\sigma$ is the signature of the knot, is true for 96.49% (516450) of the knots. These relationships are summarized in Figure 2.

As discussed in Section 2.2, the Khovanov homology of alternating knots is completely determined by classical invariants (the Jones polynomial $J(q)$ and signature $\sigma$).[25]  While Howie [48] and Greene [49] both gave topological characterizations of alternating knots, the resulting algorithms for identifying alternating knots are not practical to implement on our dataset. We thus content ourselves with using the Jones and Alexander polynomials, as well as the signature and $s$-invariant to obstruct knots in our dataset from being alternating. By identifying knots whose Jones or Alexander polynomials are not alternating,[26] along with

---

[25]It is natural to ask to what extent the Jones polynomial determines the signature of a knot. While there are knots with the same Jones polynomial and different signature, it is a folklore result that the Jones polynomial determines the signature mod 4. See, for instance, [47].

[26]Alternating knots are known to have Jones and Alexander polynomials which are alternating, meaning that coefficients of terms with consecutive powers alternate sign.

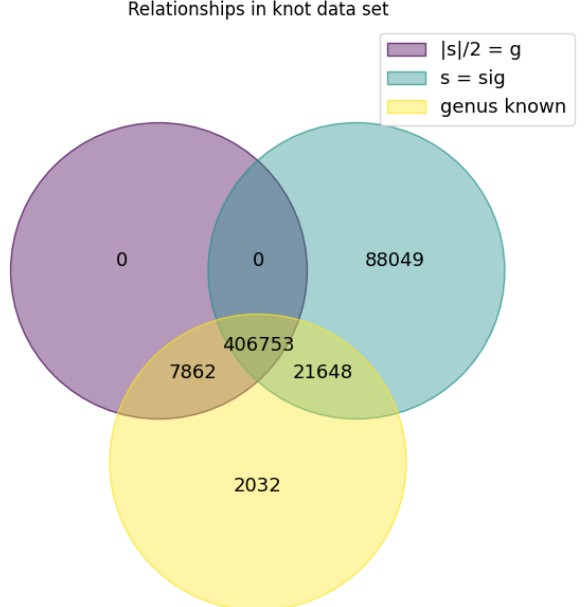

**Figure 2**: Relationships between knot invariants in the dataset. There are a total of 535239 knots, 438295 knots with known slice genus, 414615 knots where (1.3) is known to be saturated, and 497099 knots satisfying (4.3). As an example, the number 2032 represents the number of knots where we know the slice genus, but where (1.3) and (4.3) are not satisfied.

knots for which $\sigma \neq s$, we identified 117510 knots which are not alternating (approximately 21.1% of the total dataset). As none of these conditions are sufficient to imply that a knot is alternating, the remaining knots in our dataset may or may not be alternating.

## 4.3 Khovanov polynomials

In this section, we use the neural network architecture described above to learn the $s$-invariant and the slice genus from the Khovanov polynomials of the knots. The polynomials are encoded as vectors $\mathrm{Kh}(q,t) = ((e_{q_1}, e_{t_1}, c_1), (e_{q_2}, e_{t_2}, c_2), \ldots)$, where $e_{q_i}$ is the exponent of $q$, $e_{t_i}$ is the exponent of $t$, and $c_i$ is the coefficient of that term. For instance, the Khovanov polynomial for the trefoil knot,

$$\mathrm{Kh}(3_1; q, t) = q^9 t^3 + q^5 t^2 + q^3 + q, \tag{4.4}$$

is written as $(9, 3, 1, 5, 2, 1, 3, 0, 1, 1, 0, 1)$, where we have flattened the vector. The vectors are padded with zeros from the right to ensure consistently sized input. Training the neural network to predict the $s$-invariant and slice genus based on the Khovanov polynomial, we predict these invariants with 98.30% and 98.60%, respectively (averaged over five runs). The results are robust under a decrease in training size, as shown in Figure 3. When the prediction

is incorrect, we can also look at how far off the predictions are. For $s$, we look at the value $|s_{\text{true}} - s_{\text{pred}}|$. When the network incorrectly predicts $s$, it is off by two $91.42\%$ of the time, off by four $5.16\%$ of the time, and off by six $1.90\%$ of the time. For $g$, when the predictions are incorrect, they are off by one $98.56\%$ of the time and off by two $1.33\%$ of the time.

For the purpose of finding slice knots, we need only learn whether or not $g$ is zero. The network is able to successfully predict this correctly $99.26\%$ of the time, averaged over 5 runs.

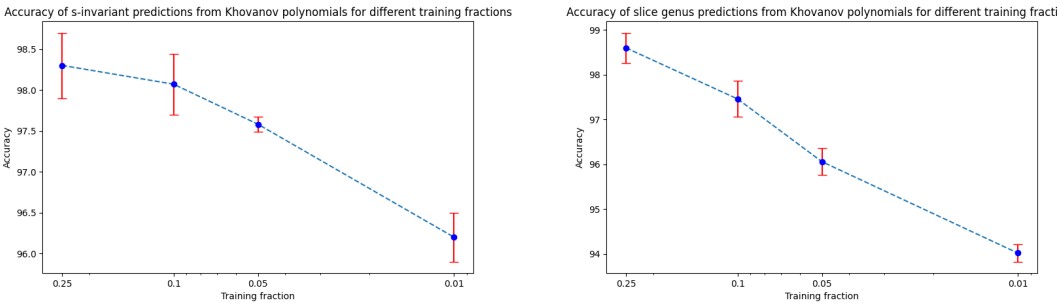

**Figure 3**: Left: Accuracy of neural network predictions of the $s$-invariant for decreasing training fractions. Right: Accuracy of slice genus predictions for decreasing training fractions. In both cases, the neural network inputs are the Khovanov polynomials, as encoded above.

### 4.3.1 Knight move experiments

The inputs here are the Khovanov polynomials $\text{Kh}(q, -q^n)$ evaluated at $t = -q^n$. The polynomials are encoded as vectors, $\text{Kh}(q, -q^n) = (e_1, c_1, e_2, c_2, \ldots)$. The $n = -4$ case corresponds to (3.7). Indeed, upon making the substitution $t = -q^{-4}$, (4.4) becomes

$$\text{Kh}(3_1; q, -q^{-4}) = q^3 + q = q^2(q + q^{-1}),\tag{4.5}$$

from which we read off $s = 2$. Table 1 shows the accuracies of the neural network in predicting the $s$-invariant and the slice genus for $|n| \leq 5$. The mean and variance are computed over five runs.

The excellent results for $n = -4$ are expected because they agree with the conjecture in (3.7), which all of the knots in the dataset satisfy. A potential source of misidentifications may stem from ambiguities in how the Khovanov polynomial is factored. We could in principle cast the Laurent expansion another way so that in addition to (2.9), we have

$$\text{Kh}(q, t) = q^{s'}(q + q^{-1}) + \sum_{\ell \geq 1} \widetilde{f}_{2\ell}(q, t)(1 + tq^{4\ell}),\tag{4.6}$$

| $n$ | $s$-invariant accuracy | slice genus accuracy | $n$ | $s$-invariant accuracy | slice genus accuracy |
|---|---|---|---|---|---|
| $-5$ | $0.9567 \pm 0.0024$ | $0.9700 \pm 0.0021$ | 5 | $0.9377 \pm 0.0114$ | $0.9633 \pm 0.0040$ |
| $-4$ | $0.9977 \pm 0.0010$ | $0.9452 \pm 0.0007$ | 4 | $0.9457 \pm 0.0028$ | $0.9652 \pm 0.0037$ |
| $-3$ | $0.9791 \pm 0.0043$ | $0.9716 \pm 0.0068$ | 3 | $0.9554 \pm 0.0019$ | $0.9656 \pm 0.0066$ |
| $-2$ | $0.9988 \pm 0.0005$ | $0.9456 \pm 0.0002$ | 2 | $0.9612 \pm 0.0013$ | $0.9663 \pm 0.0040$ |
| $-1$ | $0.9771 \pm 0.0054$ | $0.9751 \pm 0.0051$ | 1 | $0.9577 \pm 0.0033$ | $0.9765 \pm 0.0011$ |
| 0 | $0.9480 \pm 0.0021$ | $0.9720 \pm 0.0016$ | $-$ | $-$ | $-$ |

**Table 1**: Accuracy of neural network predictions for the Rasmussen $s$-invariant and the slice genus from the Khovanov polynomial evaluated at $t = -q^n$, $n \in [-5, 5]$; $\mathrm{Kh}(q, -q^n)$. The $n = -4$ case corresponds to (3.7) and $n = 0$ reduces the Khovanov polynomial to the Jones polynomial. High accuracies are achieved for $n = -4$ and $n = -2$.

with $s \neq s'$.[27] If this is what happens, the effect must be subtle. Picking out the first and second summands only, we have computed the $t = -q^{-4}$ and $t = -q^{-8}$ specializations to test whether these can give rise to different $s$ values. This does occur, but only for the unknot. Our specializations are not able to detect $s$ values that originate from knight moves of mixed sizes, however.

The $n = -2$ results, which are comparable to, and sometimes better than, the $n = -4$ results are more surprising. Table 2 demonstrates why the accuracy may be so high: we see that a majority of knots at a certain $s$-invariant have the same polynomial $\mathrm{Kh}(q, -q^{-2})$.

For all knots in the table, the polynomials evaluate to $\pm 2$ at $q = \pm 1$. The Khovanov polynomial admits an expansion (2.9). Putting $t = -q^{-2}$, this becomes

$$\mathrm{Kh}(q, -q^{-2}) = q^s(q + q^{-1}) + \sum_{\ell \geq 1} f_{2\ell}(q, -q^{-2})(1 - q^{4\ell - 2}). \tag{4.7}$$

The summands vanish for $q = \pm 1$, and indeed the $q^s = 1$ term is multiplied by $\pm 2$. According to the knight move conjecture, which is false, $f_{2\ell} = 0$ for all $\ell \geq 2$. We can express the most common polynomial for $s = -6, -4, -2, 0$ (and the second most common for $s = -8$ and $s = -10$) as

$$\mathrm{Kh}(q, -q^{-2}) = q^s \left( -\left(\tfrac{s}{2} - 1\right) q + \left(\tfrac{s}{2} + 1\right) q^{-1} \right). \tag{4.8}$$

Note that there are some cases where making this substitution produces the second most common polynomial for $s$, rather than the most common. The second most common polynomial, for $s = 0, -6$, as well as the first most common polynomial for $s = -4, -8, -10, -12$

---

[27]Though no example is stated, this possibility was already noted in [11].

| $s$ | total count | most common $\mathrm{Kh}(q,-q^{-2})$ | tally | second most common $\mathrm{Kh}(q,-q^{-2})$ | tally |
|---|---|---|---|---|---|
| $-18$ | 12 | $-q^{-19}+q^{-17}-2q^{-15}+3q^{-13}+q^{-11}$ | 4 | $-q^{-19}+q^{-17}-3q^{-15}+5q^{-13}$ | 4 |
| $-16$ | 28 | $-q^{-17}+q^{-15}-2q^{-13}+4q^{-11}$ | 19 | $-q^{-17}+3q^{-11}$ | 6 |
| $-14$ | 66 | $-q^{-15}+q^{-11}+2q^{-9}$ | 37 | $-q^{-15}+q^{-13}-q^{-11}+3q^{-9}$ | 17 |
| $-12$ | 264 | $-q^{-13}-q^{-11}+4q^{-9}$ | 151 | $-q^{-13}+q^{-7}+2q^{-9}$ | 64 |
| $-10$ | 1173 | $-q^{-11}+3q^{-7}$ | 989 | $-4q^{-11}+6q^{-9}$ | 154 |
| $-8$ | 4549 | $-q^{-9}+q^{-7}+2q^{-5}$ | 2304 | $-3q^{-9}+5q^{-7}$ | 2223 |
| $-6$ | 15075 | $-2q^{-7}+4q^{-5}$ | 11259 | $-q^{-7}+2q^{-5}+q^{-3}$ | 3775 |
| $-4$ | 32621 | $-q^{-5}+3q^{-3}$ | 32582 | $q^{-3}+q^{-1}$ | 16 |
| $-2$ | 57591 | $2q^{-1}$ | 57302 | $-q^{-3}+4q^{-1}-q$ | 2261 |
| $0$ | 339140 | $q+q^{-1}$ | 339081 | $-q^{-1}+5q-2q^{3}$ | 15 |
| $2$ | 52440 | $2q$ | 52175 | $-q^{-1}+4q-q^{3}$ | 235 |
| $4$ | 23009 | $3q^{3}-q^{5}$ | 22973 | $q+q^{3}$ | 17 |
| $6$ | 7464 | $4q^{5}-2q^{7}$ | 5361 | $q^{3}+2q^{5}-q^{7}$ | 2057 |
| $8$ | 1214 | $5q^{7}-3q^{9}$ | 604 | $2q^{5}+q^{7}-q^{9}$ | 593 |
| $10$ | 352 | $3q^{7}-q^{11}$ | 311 | $6q^{9}-4q^{11}$ | 22 |
| $12$ | 140 | $-q^{13}-q^{11}+4q^{9}$ | 63 | $-q^{13}+2q^{9}+q^{7}$ | 40 |
| $14$ | 62 | $-q^{15}-q^{11}+q^{13}+3q^{9}$ | 30 | $-q^{15}+q^{11}+2q^{9}$ | 18 |
| $16$ | 30 | $-q^{17}+q^{15}-2q^{13}+4q^{11}$ | 20 | $-q^{17}+3q^{11}$ | 7 |
| $18$ | 7 | $2q^{11}+q^{13}-q^{15}+q^{17}-q^{19}$ | 3 | $3q^{13}-2q^{15}+q^{17}+q^{11}-q^{19}$ | 2 |
| $20$ | 2 | $3q^{13}-q^{17}+q^{19}-q^{21}$ | 2 | – | – |
| – | 535239 | – | 525270 | – | 11526 |

**Table 2**: Most common Khovanov polynomials, $\mathrm{Kh}(q,-q^{-2})$, for each $s$-invariant. The neural network achieves over 99% accuracy when predicting the $s$-invariant from $\mathrm{Kh}(q,-q^{-2})$.

can be expressed as

$$\mathrm{Kh}(q,-q^{-2}) = -q^{s-1} + \left(5+\tfrac{s}{2}\right)q^{s+1} - \left(2+\tfrac{s}{2}\right)q^{s+3}. \tag{4.9}$$

This expression applies for $s \leq 0$. We obtain an expression for positive values of $s$ by sending $q \to q^{-1}$. There also appear to be related expressions for the cases where $\mathrm{Kh}(q,-q^{-2})$ has four or five terms.

At least one aspect of the simple polynomials (4.8) and (4.9) can be explained by considering a simple change of variables in the Khovanov polynomial. The key observation is that sending $t \to tq^{-2}$ leaves the leading term in the decomposition (2.9) unchanged, since $q^{s}(q+q^{-1})$ is independent of $t$. This transformation acts in a predictable way on a large class of knots with bounded homological width. In essence, it normalizes the powers of $q$ appearing in the Khovanov polynomial around those appearing at $t^{0}$. When the replacement $t = -q^{-2}$ is made, the resulting Khovanov polynomial reduces to alternating sums of terms with a fixed set of quantum gradings centered on $s$. For knots with homological width two, for instance,

| | −7 | −6 | −5 | −4 | −3 | −2 | −1 | 0 | 1 | 2 |
|---|---|---|---|---|---|---|---|---|---|---|
| 1 | | | | | | | | | | 1 |
| −1 | | | | | | | | | 1 | |
| −3 | | | | | | | | (2+1) | 1 | |
| −5 | | | | | | | 3 | (1+1) | | |
| −7 | | | | | | 4 | 2 | | | |
| −9 | | | | | 3 | 3 | | | | |
| −11 | | | | 3 | 4 | | | | | |
| −13 | | | 2 | 3 | | | | | | |
| −15 | | 1 | 3 | | | | | | | |
| −17 | | 2 | | | | | | | | |
| −19 | 1 | | | | | | | | | |

**Table 3**: The Khovanov homology of $K = 9_{20}$, represented in tabular form. Here the vertical axis represents the quantum grading, while the horizontal axis represents the homological grading. We have highlighted the single pawn move corresponding to the $q^{-4}(q+q^{-1})$ term in (2.9) at homological grading 0.

the only powers which can appear in these alternating sums are $q^{s+1}$ and $q^{s-1}$ as these are the powers appearing at $t^0$. For knots with homological width three, the relevant powers are $q^{s+3}$, $q^{s+1}$, and $q^{s-1}$. So, this argument explains the number of distinct powers appearing in the polynomial $\mathrm{Kh}(q, -q^{-2})$ as well as their relation to $s$.

To illustrate this, consider a homologically thin knot, for example $K = 9_{20}$, with Khovanov polynomial

$$
\begin{aligned}
\mathrm{Kh}(q, t) = & \frac{1}{q^{19}t^7} + \frac{2}{q^{17}t^6} + \frac{1}{q^{15}t^6} + \frac{3}{q^{15}t^5} + \frac{2}{q^{13}t^5} + \frac{3}{q^{13}t^4} + \frac{3}{q^{11}t^4} + \\
& + \frac{4}{q^{11}t^3} + \frac{3}{q^9 t^3} + \frac{3}{q^9 t^2} + \frac{4}{q^7 t^2} + \frac{2}{q^7 t} + \frac{3}{q^5 t} + \frac{2}{q^5} + \frac{3}{q^3} + \frac{t}{q^3} + \frac{t}{q} + qt^2 \, .
\end{aligned}
\tag{4.10}
$$

The polynomial $\mathrm{Kh}(q, t)$ can be represented in table form as shown in Table 3. Notice that $\mathrm{Kh}(q, t)$ can be decomposed as in (2.9) as

$$
\mathrm{Kh}(q, t) = q^{-4}(q + q^{-1}) + f_2(q, t)(1 + tq^4) \, ,
\tag{4.11}
$$

where

$$
f_2(q, t) = \frac{1}{q^{19}t^7} + \frac{2}{q^{17}t^6} + \frac{3}{q^{15}t^5} + \frac{3}{q^{13}t^4} + \frac{4}{q^{11}t^3} + \frac{3}{q^9 t^2} + \frac{2}{q^7 t} + \frac{1}{q^5} + \frac{t}{q^3} \, .
\tag{4.12}
$$

Expanding out the product $f_2(q, t)(1 + tq^4)$ above, we see that $\mathrm{Kh}(q, t)$ contains terms of the form $\alpha q^m t^n (1 + tq^4)$, for integers $m, n$, and $\alpha$, with $\alpha \geq 1$, each of which can be represented as

| | $n$ | $n+1$ |
|---|---|---|
| $m+4$ | | $\alpha$ |
| $m+2$ | | |
| $m$ | $\alpha$ | |

| | $0$ |
|---|---|
| $s+1$ | $1$ |
| $s-1$ | $1$ |

**Table 4**: Left: Knight move corresponding to the term $\alpha q^m t^n (1 + tq^4)$. Right: Pawn move corresponding to the term $q^s(q + q^{-1})$.

| | $-7$ | $-6$ | $-5$ | $-4$ | $-3$ | $-2$ | $-1$ | $0$ | $1$ | $2$ |
|---|---|---|---|---|---|---|---|---|---|---|
| $-3$ | | $1$ | $2$ | $3$ | $3$ | $4$ | $3$ | $(2+1)$ | $1$ | $1$ |
| $-5$ | $1$ | $2$ | $3$ | $3$ | $4$ | $3$ | $2$ | $(1+1)$ | $1$ | |

**Table 5**: A tabular representation of $\mathrm{Kh}(q, tq^{-2})$, where each diagonal line in $\mathrm{Kh}(q,t)$ has been flattened to a horizontal line. Notice that the terms at homological degree zero remain fixed.

a knight move as shown in Table 4 (left). The only remaining term in $\mathrm{Kh}(q,t)$ is $q^{-4}(q+q^{-1})$, which represents a lone pawn move in Table 4 (right). Notice that the location of the pawn move for $9_{20}$ at quantum gradings $-3$ and $-5$ indicates that $s = -4$ for this knot.

We may make the substitution $t \to -q^{-2}$ in two stages. First we make the substitution $t \to tq^{-2}$. This shifts the quantum grading of every term down by $-2$ times its homological grading, which has the effect of flattening each diagonal line in Table 3 into a horizontal line as in Table 5. Note that the terms at homological grading $0$ remain fixed when making this substitution. Furthermore, since (aside from lone pawn move pair) the terms of $\mathrm{Kh}(q,t)$ are arranged in knight move pairs, after substituting $t \to tq^{-2}$ we obtain two identical lines (again ignoring the pawn move pair) that are offset by one.

In the new polynomial $\mathrm{Kh}(q, tq^{-2})$, we may now set $t = -1$ to implement our desired substitution $t = -q^{-2}$ in the original polynomial $\mathrm{Kh}(q,t)$. This step may be interpreted as taking the alternating sum of the rows of Table 5, and multiplying by $q^{-3}$ and $q^{-5}$ respectively. Since the rows are offset by one, the alternating sums will differ by a sign until the pawn move pair is factored in ($2$ and $-2$ respectively), which then shifts both sums up by one. In this case we obtain

$$\mathrm{Kh}(q, -q^{-2}) = -q^{-5} + 3q^{-3}, \tag{4.13}$$

which matches (4.8) for $s = -4$. A similar argument may be used to partially explain (4.9).

Notice that although this explains the general form that the polynomials $\mathrm{Kh}(q, -q^{-2})$ often take, it does not explain why the coefficients of these terms seem to also be completely fixed by the $s$-invariant. The path through Lee homology taken by Rasmussen in order to define the $s$-invariant makes reference only to the remaining quantum grading since Lee homology is isomorphic to $\mathbb{Q} \oplus \mathbb{Q}$ at homological degree 0. Our observation suggests that perhaps $s$ is encoded in some additional way within Khovanov homology, one which does not make reference to Lee homology or the quantum grading.

### 4.3.2 Learning from polynomial evaluations

We can also learn the $s$-invariant from the polynomials $\mathrm{Kh}(q, -q^{-4})$ and $\mathrm{Kh}(q, -q^{-2})$, evaluated at roots of unity $e^{\frac{\pi i n}{(k+2)}}$ for $n \in [0, k+2)$. The input data for a knot $K$ is represented as a vector $\vec{v}(K; k)$, where the entries at position $2p$ and $2p+1$ correspond to the real and complex parts of the $p$-th evaluation. For instance, taking $k = 3$, the Khovanov polynomial for the trefoil knot at $t = -q^{-4}$, which we write in (4.5), is encoded as

$$\vec{v}(3_1; 3) = (2, 0, 0.5, 1.54, -0.5, 0.36, 0.5, 0.36, -0.5, 1.54) . \qquad (4.14)$$

When training on either evaluations of $\mathrm{Kh}(q, -q^{-4})$ or of $\mathrm{Kh}(q, -q^{-2})$, the accuracy of predictions is over 99% for the $s$-invariant and over 94% for the slice genus, even for $k = 3$, where we train on only five evaluations of the polynomial.

## 4.4 Jones polynomials

The Jones polynomials are encoded, like the Khovanov polynomials, as vectors, $J(q) = (e_1, c_1, e_2, c_2, \ldots)$. Using this input and training on 25% of the data, the network predicts the Rasmussen $s$-invariant with $\sim 95.0\%$ accuracy, and the slice genus with $\sim 96.6\%$ accuracy. The results for the $s$-invariant are fairly robust under a decrease in training size, Figure 4. When the prediction is incorrect, we can also look at how far off the predictions are. For $s$, we look at the value $|s_{\mathrm{true}} - s_{\mathrm{pred}}|$. When the network incorrectly predicts $s$, it is off by two 67.83% of the time, off by four 18.28% of the time. For $g$, when the predictions are incorrect, they are off by one 81.71% of the time and off by two 17.52% of the time. The accuracy of these networks is surprising, as there are knots with the same Jones polynomial but different slice genus and $s$-invariant.[28] This means that the neural network is not learning a function, but merely an association.

For the purpose of finding slice knots, we need only learn whether or not $g$ is zero. The network is able to successfully predict this correctly 98.61% of the time, averaged over 5 runs.

---

[28]The knots $5_1$ and $10_{132}$ are a pair of such knots.

| $k$ | $s$-invariant accuracy | Slice genus accuracy |
|---|---|---|
| 3 | $0.9314 \pm 0.0013$ | $0.9672 \pm 0.0017$ |
| 4 | $0.9186 \pm 0.0009$ | $0.9601 \pm 0.0010$ |
| 5 | $0.9650 \pm 0.0006$ | $0.9826 \pm 0.0009$ |
| 6 | $0.9674 \pm 0.0007$ | $0.9828 \pm 0.0008$ |
| 7 | $0.9676 \pm 0.0007$ | $0.9825 \pm 0.0008$ |
| 8 | $0.9673 \pm 0.0005$ | $0.9825 \pm 0.0006$ |
| 9 | $0.9669 \pm 0.0011$ | $0.9814 \pm 0.0014$ |
| 10 | $0.9680 \pm 0.0006$ | $0.9826 \pm 0.0008$ |

**Table 6**: Learning the Rasmussen $s$-invariant and the slice genus from evaluations of the Jones polynomial at roots of unity, $e^{\pi in/(k+2)}$ for $n \in [0, k+2)$. This experiment is repeated for $k \in [3, 10]$.

We can also learn the invariants from evaluations of the Jones polynomial at roots of unity.

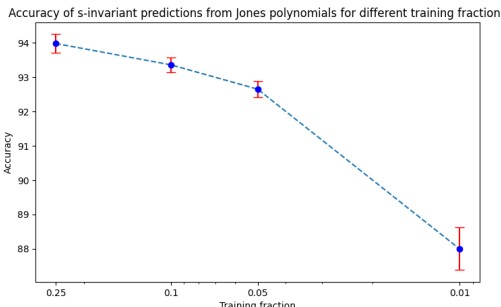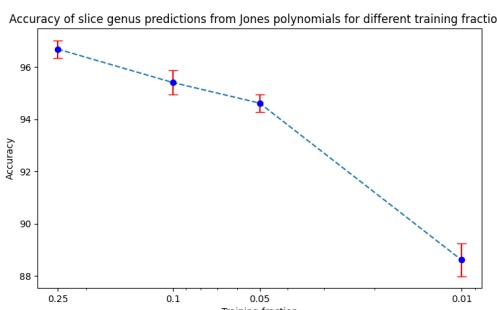

**Figure 4**: Left: Accuracy of neural network predictions of the $s$-invariant for decreasing training fractions. Right: Accuracy of slice genus predictions for decreasing training fractions. In both cases, the neural network inputs are the Jones polynomials, as encoded above.

For a fixed $k$, we generate a list of roots $\{e^{\pi in/(k+2)}\}$, $n \in [0, k+2)$ at which to evaluate the Jones polynomials. The results for training the network on these inputs, for $k \in [3, 10]$, are given in Table 6.

## 4.5 Interdependence of results

As discussed previously, the Rasmussen $s$-invariant provides a lower bound for the slice genus. Our neural networks are able to successfully predict both $s$ and $g$, but we do not know if the two invariants are learnt independently or if one is learnt via the other. Additionally, since the signature and the $s$-invariant coincide for most of the knots, it could be that the network

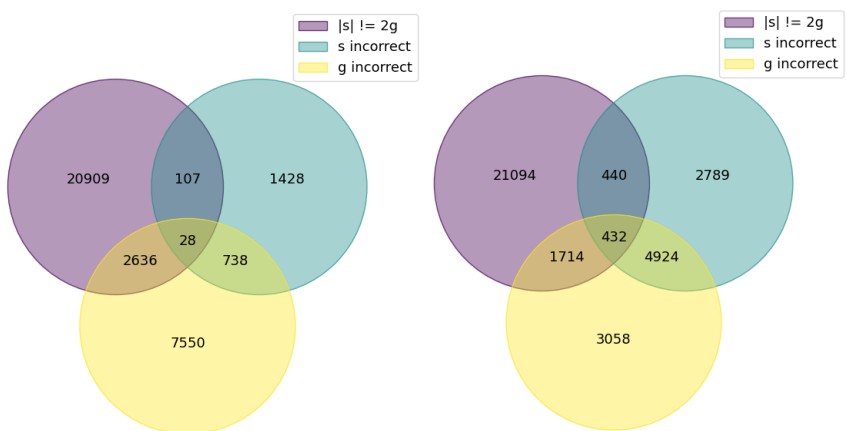

**Figure 5**: Comparing the number of $s$-invariants and slice genuses predicted incorrectly by neural networks with the number of cases where the slice genus saturates the lower bound provided by the $s$-invariant. Left figure is trained on the Khovanov polynomial and right figure is trained on the Jones polynomial.

is learning $s$ indirectly via the signature. In this section, we aim to diagnose which invariants the neural networks are learning directly.

First, we train two neural networks to learn the $s$-invariant and the slice genus, using the Khovanov (or Jones) polynomials. Then, we count how many times $s$ and $g$ were predicted incorrectly when $|s| = 2g$, versus when $|s| \neq 2g$. These counts are summarized in Figure 5. When training on the Khovanov polynomial (Figure 5 left), a total of 2301 $s$-invariants and 10952 slice genuses are predicted incorrectly. The majority of the $s$-invariants which are predicted incorrectly lie in the region where $|s| = 2g$. This does not support the claim that the network is learning $s$ via $g$. A similar trend is apparant when training on the Jones polynomials (Figure 5 right).

We can perform a similar experiment with the signature and the $s$-invariant. The results for this experiment are given in Figure 6. When training on the Khovanov polynomial (Figure 6 left) a total of 7438 $s$-invariants and 4280 signatures were predicted incorrectly. When $s$ and $\sigma$ coincide, $s$ is predicted incorrectly 1648 times and $\sigma$ is predicted incorrectly 6693 times. When $s$ and $\sigma$ do not coincide, $s$ is predicted incorrectly 3510 times and $\sigma$ is predicted incorrectly 2939 times. This experiment does not offer any conclusive evidence as to which invariant the network is learning.

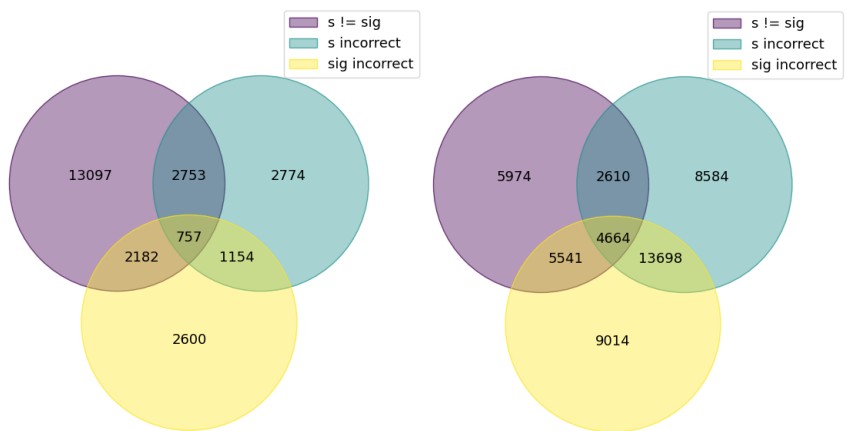

**Figure 6**: Comparing the number of $s$-invariants and signatures predicted incorrectly by a neural network trained on Khovanov polynomials with the number of cases where $s = \sigma$. Left figure is trained on the Khovanov polynomials and right figure is trained on the Jones polynomials.

### 4.5.1 Multi-task learning

We can further train a neural network to learn the $s$-invariant and the slice genus simultaneously. This may provide insight into which invariant the network is learning. In a multi-task problem, we can weight the various tasks. This weight determines how strongly that task will influence the change of parameters as the network is trained. We can ask how changing the weights on the two outputs affects the overall learning of the network. Denote the weight of the $s$-invariants to be $\mathcal{W}_s$ and the weights of the slice genus to be $\mathcal{W}_g$. We find that weighting either the $s$-invariant or the slice genus more heavily did not significantly influence the performance of the network on either of the invariants.

First, we try this experiment using the Khovanov polynomials as input. For $\mathcal{W}_s/\mathcal{W}_g = 1$, we get accuracies of $0.9952 \pm 0.0002$ and $0.9866 \pm 0.0026$ on the $s$-invariant and the slice genus, respectively. When $\mathcal{W}_s/\mathcal{W}_g = 0.1$, we get accuracies of $0.9924 \pm 0.0010$ and $0.9859 \pm 0.0044$. For $\mathcal{W}_s/\mathcal{W}_g = 10$, we get accuracies of $0.9954 \pm 0.0004$ and $0.9797 \pm 0.0025$. Repeating this experiment only on the dataset where $|s| \neq 2g$, we get the following results. For $\mathcal{W}_s/\mathcal{W}_g = 1$, we get accuracies of $0.9959 \pm 0.0002$ and $0.9943 \pm 0.0008$. When $\mathcal{W}_s/\mathcal{W}_g = 0.1$, we get accuracies of $0.9954 \pm 0.0022$ and $0.9935 \pm 0.0027$. For $\mathcal{W}_s/\mathcal{W}_g = 10$, we get accuracies of $0.9960 \pm 0.0003$ and $0.9948 \pm 0.0008$.

We can also do this experiment using the Jones polynomials as input. For $\mathcal{W}_s/\mathcal{W}_g = 1$, we get accuracies of $0.9775 \pm 0.0007$ and $0.9723 \pm 0.0043$ on the $s$-invariant and the slice genus, respectively. When $\mathcal{W}_s/\mathcal{W}_g = 0.1$, we get accuracies of $0.9766 \pm 0.0006$ and $0.9731 \pm 0.0032$. For $\mathcal{W}_s/\mathcal{W}_g = 10$, we get accuracies of $0.9774 \pm 0.0021$ and $0.9640 \pm 0.0084$. Repeating this

experiment only on the dataset where $|s| \neq 2g$, we get the following results. For $\mathcal{W}_s/\mathcal{W}_g = 1$, we get accuracies of $0.9803 \pm 0.0006$ and $0.9786 \pm 0.0021$. When $\mathcal{W}_s/\mathcal{W}_g = 0.1$, we get accuracies of $0.9806 \pm 0.0008$ and $0.9788 \pm 0.0010$. For $\mathcal{W}_s/\mathcal{W}_g = 10$, we get accuracies of $0.9811 \pm 0.0006$ and $0.9780 \pm 0.0016$.

## 5    Discussion

In this work, we have demonstrated that the Jones polynomial of a knot is strongly correlated with the knot's Rasmussen $s$-invariant and slice genus $g$. We also found a specialization for the Khovanov polynomial which makes the $s$-invariant manifest in our dataset, though this specialization is not the one suggested by the knight move conjecture. These correlations were extracted by training deep neural networks to predict $s$ or $g$ with the Jones or specialized Khovanov polynomials as input. We reviewed both the combinatorial construction of Khovanov homology and Rasmussen's invariant as well as the gauge-theoretic constructions in four, five, and six dimensions.

Though we were unable to determine whether the neural network learned $s$, $g$, some combination of the two, or something else altogether,[29] there is at least one naïve reason to believe that the Jones polynomial is correlated with $s$ specifically. This is because both arise from Khovanov homology, whereas the slice genus is only known through Khovanov homology relative to the $s$-invariant. The gauge theoretic construction of Khovanov homology in [7] combined with the knight move conjecture makes it clear that (in our dataset) $s$ may be extracted from the Hilbert space of a five- or six-dimensional theory with surface operators, whereas the construction of $s^\sharp$ in [30] shows that invariants like $s$ can arise in the instanton homology of simpler gauge theories.

Despite these relationships, there is no obvious route by which the Jones polynomial may be manipulated into revealing the $s$-invariant. A potential avenue, motivated by the index formula for $J(q)$ in the five-dimensional theory, is that the analytic structure of $J(q)$ around $q = 1$ is sensitive to $s$ because $s$ can be extracted from the graded trace with $t = -q^{-4}$ at large $k$. However, this requires $J(q)$ to somehow be sensitive to analytic structure in the $t$ variable of the Khovanov polynomial. It is not clear why this would be so, as $J(q)$ is obtained from $\mathrm{Kh}(q,t)$ by immediately taking $t = -1$. These results could imply that Chern–Simons gauge theory knows a little bit more about the five-dimensional gauge theory Hilbert space than is naïvely expected. In a previous situation where Chern–Simons theory appeared to contain

---

[29]A major obstruction to answering this question was that the vast majority of our dataset obeyed $|s| = 2g$. A useful step forward would be to find a knot generation procedure which yields many knots for which $|s| < 2g$.

more information than it should [14], the explanation was found in an analytic continuation of the path integral [16]. Perhaps the same effect is at work here. Another approach which may be useful begins with the observation that there are spectral sequences relating Khovanov homology to *e.g.*, instanton Floer homology [50, 51]. The gauge theory interpretation of these spectral sequences is unclear, but finding such an interpretation may help in relating $J(q)$ to $s$, since the $s$-like invariant $s^\sharp$ is easily extracted from instanton Floer homology. The question concerning the four-dimensional information in $J(q)$ that we posed in the introduction still stands, and a complementary question is: what is three-dimensional about the Rasmussen $s$-invariant?

The specialization of the Khovanov polynomial at $t = -q^{-2}$, on the other hand, had at least a partial explanation via the knight move conjecture for knots in our dataset. This explanation was given in terms of a rearrangement of the Khovanov homology groups into a number of rows determined by the homological width of the knot. This width determined the number of nonzero terms (in the most general case with no cancellations), and the powers of these terms had an offset which depended on $s$. However, we were not able to explain why the coefficients of the resulting polynomial also seem to be determined completely by $s$. This suggests that the $s$-invariant may be extracted from Khovanov homology in a way that differs from the route through Lee homology taken by Rasmussen.

The fact that machine learning is successful at identifying relationships between knot invariants is suggestive, both with respect to low dimensional topology and to quantum field theory. Because knot invariants have been extensively tabulated [41, 44], one could easily imagine using neural networks to scan through all pairs of invariants and determine which can be predicted from knowledge of the other. As we do not have a minimal list of topological invariants that identifies a knot uniquely, such an investigation may provide some guidance in this endeavor. We have also seen that knot invariants have a physical interpretation in diverse dimensions. The Jones polynomial, for example, can be thought of as a mathematical object with a two-dimensional definition or as a physical quantity associated to quantum field theories in three, four, five, or six dimensions. Translating the relationships between knot invariants to a physical language may expose non-obvious connections between quantum field theories. As an example of this approach, the knot-quiver correspondence [52] may be helpful to more generally organize relationships between physical knot invariants.[30]

For instance, in the five-dimensional theory discussed in Section 3, the formula we presented for $s$ holds for a certain subset of knots. What this formula means is that there exists a very special pair of approximate supersymmetric ground states in the five-dimensional theory

---

[30]We thank Sergei Gukov for bringing this correspondence to our attention.

which have instanton numbers equal to $s \pm 1$. These states survive the five-dimensional instanton corrections to the ground state energies. Our results suggest that the Chern–Simons path integral has privileged access to these states in particular, through some yet undiscovered gauge theory mechanism. Furthermore, the information contained in Lee homology and the specialization $\mathrm{Kh}(q, -q^{-2})$ about the $s$-invariant may be interpreted as a statement about the existence of deformations of the topological supercharge $Q$. The nontrivial prediction from Lee's theorem in this context would be that the cohomology of a deformed supercharge is two-dimensional for any knot, and our results on the unusual specialization $\mathrm{Kh}(q, -q^{-2})$ suggest that correlating the fermion number grading with the instanton number grading in this way produces, $e.g.$, a number of ground states with instanton numbers that depend on $s$ in the manner we described in Section 4. Perhaps the perspective on deformations of supercharges via spectral sequences presented in [53] would be useful here.[31]

Finally, as poetically and playfully contemplated by the title of [11], machine learning may fill an important niche in constructing potential counterexamples to SPC4. A counterexample establishes the existence of exotic smooth structures on manifolds that are homeomorphic to $S^4$. If we can conclude whether a knot is slice from a description of the knot ($e.g.$, by using a picture or the braid representation or a collection of other topological invariants), this may highlight interesting knots to examine using analytical methods. Since we are only interested in distinguishing $g = 0$ from $g \neq 0$, this is a conceptually simpler task than learning the slice genus exactly as we have aimed to do from the Khovanov and Jones polynomials. Applying a neural network trained to identify sliceness on these knots is a promising attack for the future.

### Acknowledgments

We thank Dror Bar-Natan, Sergei Gukov, Seungwon Kim, Scott Morrison, Dirk Schütz, Jackson Switzer, and Edward Witten for helpful discussions during the preparation of this paper. JC and VJ are supported by the South African Research Chairs Initiative of the Department of Science and Technology and the National Research Foundation. VJ is additionally supported by a Simons Foundation Mathematics and Physical Sciences Targeted Grant, 509116. AK is supported by the Simons Foundation through the It from Qubit Collaboration.

---

[31]However, it is difficult to relate the two-dimensional perspective of [53] with the five-dimensional picture of [7]. In particular, while the two-dimensional construction seems to suggest the explicit symmetry breaking pattern $\mathfrak{sl}(2) \rightarrow \mathfrak{sl}(1) \oplus \mathfrak{sl}(1)$ (implemented by modifying the superpotential) is important to obtain Lee homology, this may not be the relevant mechanism in five dimensions to implement Lee's deformation. We thank Edward Witten for pointing this out to us.

# A  Implementation

In this appendix, we supply the key part of our implementation. Listing 1 outlines the building and training of the neural network.

Listing 1: Keras implementation

```python
import numpy as np
import keras
from sklearn.model_selection import train_test_split

train_size = 0.25  # training fraction
epochs = 50  # number of epochs to train for
density = 100  # density of hidden layers in network
activation = 'relu'  # activation function for input/hidden layers
loss = 'sparse_categorical_crossentropy'  # loss function for multi-label
    classification
optim = 'adam'  # optimizer

# parameters: full set of inputs and outputs
# returns: trained model and testing data (input, output)
def learn(inputs, outputs):
    # fix input dimensions
    if np.isscalar(inputs[0]):
        dims = 1
    else:
        dims = len(train_in[0])

    # calculate number of output classes
    num_classes = max(outputs) - min(outputs) + 1

    # split into training/testing data
    train_in, test_in, train_out, test_out = train_test_split(inputs, outputs,
                                           train_size=train_size)

    # build the model
    model = keras.models.Sequential([
        keras.layers.Dense(density, activation=activation, input_dims=dims),
        keras.layers.Dense(density, activation=activation),
        keras.layers.Dense(num_classes, activation='softmax'
        )]

    # compile and train the model
    model.compile(optimizer=optim, loss=loss, metrics=['accuracy']
    model.fit(train_in, train_out, epochs=epochs, verbose=0.5)
```

```
39
40        return model , test_in , test_out
```

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
