# Peer review of "Learning knot invariants across dimensions"

_SciPost Physics_

## Round 1 · Referee Report · Anonymous (Referee 1) · 2022-7-8

Report

Dear editor,

This is a nice paper and I would recommend publication with minor revision.

Here are some comments:

Introduction is well written and includes good motivation as to why it is worthwhile studying the relationships between knot invariants in terms of working towards a dimension independent theory of knots.

Section 2 introducing Khovanov homology is clear and given at an appropriate level of detail.
W on page 5 is not defined. Is W in the definition of W{k} supposed to be V?
Section 3 describes the physical interpretation of the knot invariants in terms of gauge theory. This

section is well written and connection of the results to this is made in the discussion section.
In section 3.5 the s# invariant is introduced which is not equal to the s invariant. Could you use this work to deduce some relationship between these two? For instance perform multi-task learning like in Section 4.5.1.

Results are presented well in Section 4 and their interpretation discussed critically.
Using the same y-axis range in the plots in Figure 3 would make it easier to compare the two and again for plots in Figure 4.

In section 4.5.1 please add a short explanation to make it clear how the results of differing the weights for the two invariants tells us what the network is learning.

one typo in the first paragraph of section 2.1 on line 4 where they repeated ‘for for’.

Authors are commended for including their code script in the appendix. A link to a GitHub repo would also be helpful for readers.
  • validity: good
  • significance: good
  • originality: good
  • clarity: high
  • formatting: perfect
  • grammar: excellent

Author:  Jessica Craven  on 2022-10-26  [id 2951]

(in reply to Report 1 on 2022-07-08)

We are grateful for the suggested improvements to the text. We have included a list of changes along with our resubmission.

The suggestion of learning $s^\sharp$ is excellent, but we do not have sufficiently many examples of knots with different values of $s$ and $s^\sharp$ to perform the experiment. While the Rasmussen $s$-invariant is often readily computable, the Kronheimer and Mrowka $s^\sharp$-invariant is not. Indeed, the argument that these differ even for the trefoil is subtle and was originally missed in the mathematics literature.

---

## Round 1 · Referee Report · Anonymous (Referee 2) · 2022-7-29

Report

In this paper the authors use machine learning techniques to find relations between various knot invariants: Khovanov and Jones polynomials, s-invariant, and slice genus. It is shown that a relatively simple neural network can very well predict s-invariant and slice genus from both Khovanov and Jones polynomials. The authors argue that successful predictions based on the Khovanov polynomial can be expected in view of 4-dimensional interpretation of the objects involved (Khovanov polynomial, s-invariant and slice genus), while the prediction based on the Jones polynomial is harder to explain in view of its 3-dimensional (rather than 4-dimensional) origin. The authors briefly speculate on interpretations of these results in terms of gauge theories.

The results of the paper are interesting, although not groundbreaking. It is not surprising that recently developed powerful techniques in machine learning can be applied to knots and links, for which a lot of various invariants can be computed and have been tabulated. Nonetheless, it is interesting to combine machine learning techniques with knot theory, and developing this research direction could presumably lead to some real breakthrough, as it happened in other fields. In addition, the first part of the paper is a nice summary of various results from knot theory, which should be useful for readers who are not experts in this area, but are interested in developing this research direction. For these reasons I recommend this paper for publication.

---

## Editorial Decision

resubmitted